# Time-related immunomodulation by stressors and corticosterone transdermal application in toads

**Stefanny Christie Monteiro Titon**◉*, **Braz Titon Jr, Adriana Maria Giorgi Barsotti, Fernando Ribeiro Gomes, Vania Regina Assis**◉

Departamento de Fisiologia, Instituto de Biociências, Universidade de São Paulo, São Paulo, SP, Brasil

* stefannychristie@gmail.com

**Data Availability Statement:** All relevant data are within the paper and its Supporting Information file.

## Abstract

Immune responses have been mostly studied at a specific time in anuran species. However, time-changes related to immunomodulation associated with glucocorticoid (GC) alterations following stressors and GC treatment are complex. The present study describes time-related changes in immune response and corticosterone (CORT) plasma levels following restraint challenge, short, mid and long-term captivity, and CORT exogenous administration by transdermal application (TA) in *Rhinella ornata* toads. We observed increased neutrophil: lymphocyte ratios after restraint challenge and CORT TA, without changes following short and mid-term captivity. Plasma bacterial killing ability was sustained in all treatments, except long-term captivity, with decreased values after 90 days under such conditions. Phagocytic activity of peritoneal cells increased after mid-term captivity, and the phytohemagglutinin swelling response was impaired in those animals treated with CORT TA for 20 consecutive days. Plasma CORT levels increased or were sustained after restraint challenge (depending on initial values), decreased following mid and long-term captivity (for those animals showing high CORT in the field) and increased after 20 days of CORT TA. By performing assessments of time-changes in immune processes and CORT plasma levels in *R. ornata*, we demonstrate immuno-enhancing effects following restraint, short and mid-term stressors, while long-term stressors and CORT TA promoted immunosuppression in these toads.

## Introduction

Stress-induced immunomodulation is complex and frequently associated with immune suppressive effects. However, recent studies have shown that changes in immune responses are mostly dependent on the stress duration and intensity, and can vary over time [1,2]. While long-term stress usually results in immune suppressive effects, short-term stress can enhance the immune response [2]. Meanwhile, glucocorticoids (GCs), one of the major mediators of the stress response, influence immune cell redistribution and immune function in several vertebrates, including amphibians [3–7]. Leukocyte redistribution is associated with wound healing by increasing immune responsiveness in body compartments requiring immune protection [3,8,9]. As a result, this leukocyte redistribution is reflected as an increase in blood

**Funding:** This work was supported by Fundação de Amparo à Pesquisa do Estado de São Paulo (FAPESP) (grant numbers: 2013-00900-1, and 2014/16320-7), PhD scholarships to VRA (2011/10560-8), and AMGB (2016/05024-3). It was also supported by Brazilian Conselho Nacional de Desenvolvimento Científico e Tecnológico (CNPq) through a PhD scholarship to SCMT (142455/2013-0). F.R. Gomes is a research fellow from the Brazilian CNPq - #302308/2016-4.

**Competing interests:** The authors have declared that no competing interests exist.

neutrophil: lymphocyte ratios (NLR), a common proxy of the stress response in both field and laboratory studies [8]. Recent studies have shown increased NLR following an array of stressors and GC treatment in several vertebrates, including anurans submitted to restraint challenge and corticosterone (CORT) treatments [4,8,10–13].

Immune cellular components are strongly sensitive to stressors, being upregulated within hours by the acute stress response and GC exogenous administration [14–17]. Commonly observed immune-enhancement responses include changes in leukocyte trafficking, maturation, and augmentation of cell function, including: phagocytosis activity, and production of antibodies and cytokines [2,14,18]. On the other hand, delayed wound healing, down-regulation of adaptive immunity, and anti-inflammatory actions are more pronounced days after stress exposure and GC treatment [1,2,18,19]. Specifically in the context of regulation of the inflammatory response, the immunosuppressive effects of the GCs are important for the inflammatory resolution phase [3,5]. However, long-term stress and chronic GCs exposure and their inducing anti-inflammatory actions impair mounting of the inflammatory/innate immune response [2]. Although temporal patterns of stress-induced immunomodulation have been well-described for mammals [2], data for other vertebrates are still scarce [1].

In amphibians an array of stress protocols have been used in order to investigate stress-induced CORT changes, including restraint, captivity and exogenous CORT administration [12,20–27], with recent studies also investigating stress-induced immunomodulation [12,20–23,27]. Restraint and captivity stress-induced changes include increased CORT along with both immune- enhancing or suppressive effects in anurans, depending on the studied immune parameter, as well as variation in stress duration and intensity [11,12,21,23,27]. The NLR, for example, increased following restraint without movement restriction, but even higher values were observed after restraint with movement restriction, indicating the NLR changes are correlated with intensity of the stressor [11]. Phagocytosis activity is also up-regulated following acute restraint and CORT exogenous application [12,23]. Additionally, changes in cellular and non-cellular aspects of immune response are not synchronized following long-term captivity in toads, with down-regulation of phagocytic activity and bacterial killing ability (BKA) being observed at different time-points of captivity [27,28]. Although evidences of stress-induced immunomodulation in anurans has been provided, they are generally derived from different combinations of experimental protocols (i.e., restraint, captivity, hormonal exogenous administration), and immune parameters, such as, NLR, phagocytosis, BKA, phytohemaglutinin (PHA) swelling response, among others [4,20,21,28,29]. Moreover, immune assessment has been usually performed in each of these experiments at single sampling time-points, and the studies were conducted with different species [4,21,23,29]. The multiplicity of experimental protocols, immune parameters and species used, along with measurements performed at single time-points after treatments, limit our understanding of the temporal patterns of interaction between response to stressors, CORT and immune responses in amphibians.

In this study, we investigated time-related patterns of immunomodulation in response to short and long-term stressors (restraint and captivity) and exogenous CORT administration in a single toad species, *Rhinella ornata*. We hypothesized that exposure of toads to short-term stressors would generate immuno-enhancing effects, while exposure of toads to long-term stressors and daily CORT administration for 20 days would generate immunosuppression. In order to test these hypotheses, we measured time-related changes in CORT plasma levels, NLR, BKA, PHA swelling response and phagocytic activity from peritoneal and blood leukocytes in response to short-term restraint challenge (1 and 24 hours), short, mid and long-term captivity (7 days, 30 days, and 60 to 90 days, respectively), and CORT exogenous administration (transdermal application–TA for 20 days) in *R. ornata* toads. We predicted: (1) Increased CORT plasma levels and NLR following restraint, short and long-term captivity and CORT

TA; (2) Increased BKA and phagocytosis activity in response to short-term restraint and short-term captivity; (3) Decreased BKA, phagocytosis activity and PHA edema in response to long-term captivity maintenance and CORT TA; (4) mid-term captivity (30 days) would represent a transient period, when patterns of stress response of NLR, BKA and phagocytosis percentage (PP) would be less obvious.

## Results

Except for PHA relative edema, which was positively affected by body mass and snout-vent length ($P$ = 0.036 and $P$ < 0.001, respectively), there were no relationship between morphological measures (body mass, snout-vent length and body index) and any other studied variable at any circumstance: NLR ($P \geq$ 0.193); BKA ($P \geq$ 0.101); PP ($P \geq$ 0.062); CORT plasma levels ($P \geq$ 0.193); PHA ($P \geq$ 0.541, for body index).

Toads displaying high CORT levels in the field (those exhibiting calling behavior at the moment of capture [27]) showed a tendency to decrease CORT after restraint for 24h (S1 Table, $P$ = 0.082, Fig 1A). Meanwhile, toads characterized by low CORT levels at time 0h (before initiating restraint) showed increased CORT plasma levels after 1 and 24h post-restraint (S1 Table, $P$ = 0.003, Fig 1B). Short-term captivity showed no effect on CORT plasma levels (S1 Table, $P$ = 0.844–7 days, Fig 1C). Otherwise, for those animals exhibiting high CORT in the field, mid or long-term captivity decreased CORT plasma levels (S1 and S2 Tables, $P$ = 0.049–30 days, Fig 1D; $P \leq$ 0.001–7 to 90 days, Fig 1E). In the CORT TA experiment, toads from control and placebo groups did not differ in CORT plasma levels (S3 and S4 Tables, $P$ = 1.000). Subsequently, they were grouped for further analyses and thereafter referred to as the control group. Plasma CORT levels were 14 times higher in TA-treated toads compared to controls (S2 Table, $P \leq$ 0.001, Fig 1F).

Increased values for NLR were found following restraint challenge (S5 Table, $P$ = 0.025 – 24h, Fig 2A). No differences were observed in NLR following short and mid-term captivity (S5 Table, $P$ = 0.341–7 days, Fig 2B and $P$ = 0.540–30 days, Fig 2C). Toads from both control and experimental treatments showed increased NLR after 20 days of the CORT TA treatment ($P$ = 0.031), without treatment effects (S6 Table, Fig 2D).

Plasma BKA was not altered by restraint (S7 Table, $P$ = 0.173 – 24h, Fig 3A and $P$ = 0.787–1 and 24h, Fig 3B) or short-term captivity ($P$ = 0.500–7 days, Fig 3C). Mid-term captivity for 30 days showed no effect on BKA (S7 Table, $P$ = 0.326–30 days, Fig 3D). Meanwhile, BKA was lower on the 90th day of captivity when compared to field values (S8 Table, $P$ = 0.006, Fig 3E). CORT TA for 20 consecutive days did not affect BKA (S8 Table, $P$ = 0.648, Fig 3F).

Phagocytosis percentage (PP) was not affected by restraint for 1 or 24h (S9 Table, $P$ = 0.897, Fig 4A). Whereas, PP was affected by captivity (S9 Table, $P$ = 0.053–7 to 90 days), with the maximum mean value on the 30th day of captivity (Fig 4B).

The PHA relative edema was affected by both body mass and CORT TA treatment ($P$ = 0.010), being higher at 12h when compared to 24h for both groups (Fig 4C). However, the experimental group had lower PHA swelling at both 12 and 24h when compared with control group (S10 Table, Fig 4C). There were no differences in body mass and snout-vent length between treatment groups ($t_{17}$ = -1.245, $P$ = 0.230 and $t_{17}$ = -1.540, $P$ = 0.142, respectively).

Compiled results are schematically summarized in Table 1. Complete statistical tables regarding results described at this section are presented as supplementary material (S1 to S10 Tables).

## Discussion

Plasma CORT levels increased following restraint and CORT TA, but contrary to our prediction, decreased in response to short, mid and long-term captivity in *R. ornata* toads (Table 1).

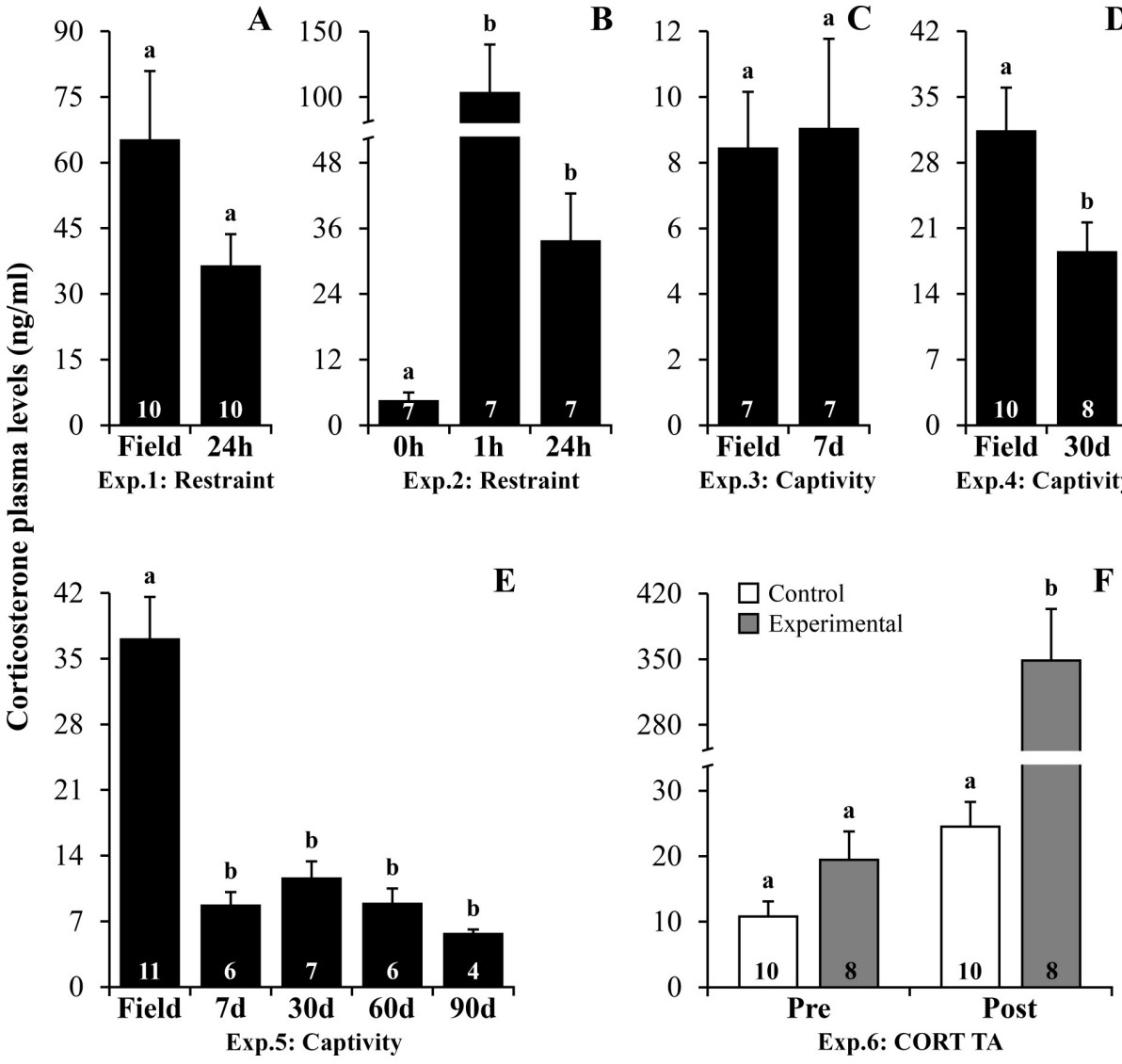

**Fig 1. Corticosterone plasma levels following stressors and corticosterone transdermal application in *R. ornata* toads.** (A) Experiment 1: Restraint–field *vs*. 24h. (B) Experiment 2: Restraint– 0h *vs*. 1h *vs*. 24h. (C) Experiment 3: Captivity–field *vs*. 7 days. (D) Experiment 4: Captivity–field *vs*. 30 days. (E) Experiment 5: Captivity–field *vs*. 7 *vs*. 30 *vs*. 60 *vs*. 90 days. (F) Experiment 6: Corticosterone transdermal application. Bars represent mean ± standard error. The N is indicated inside each bar. Letters above the bars represent statistical differences for Bonferroni adjustments in ANOVAs or Student t-test, with different letters representing statistical difference within groups with $P \leq 0.05$. Abbreviation: **Exp.:** Experiment. **CORT TA:** Corticosterone transdermal application. Legend: **Control:** control and placebo group; **Experimental:** corticosterone group.

These results indicate that restraint was perceived by these toads as a stressor, but they might be less sensitive to captivity maintenance. Moreover, CORT TA increased CORT plasma levels 1h after treatment and may have simulated the hormonal response to a repeated stressor. As we expected, NLR increased after restraint and peritoneal PP showed increased values following mid-term captivity, indicating an immune enhancing effect during this period (Table 1). Moreover, BKA and PHA edema were impaired after long-term captivity and CORT TA, which could be associated with chronic stress and GC-induced immunosuppressive effects, respectively (Table 1).

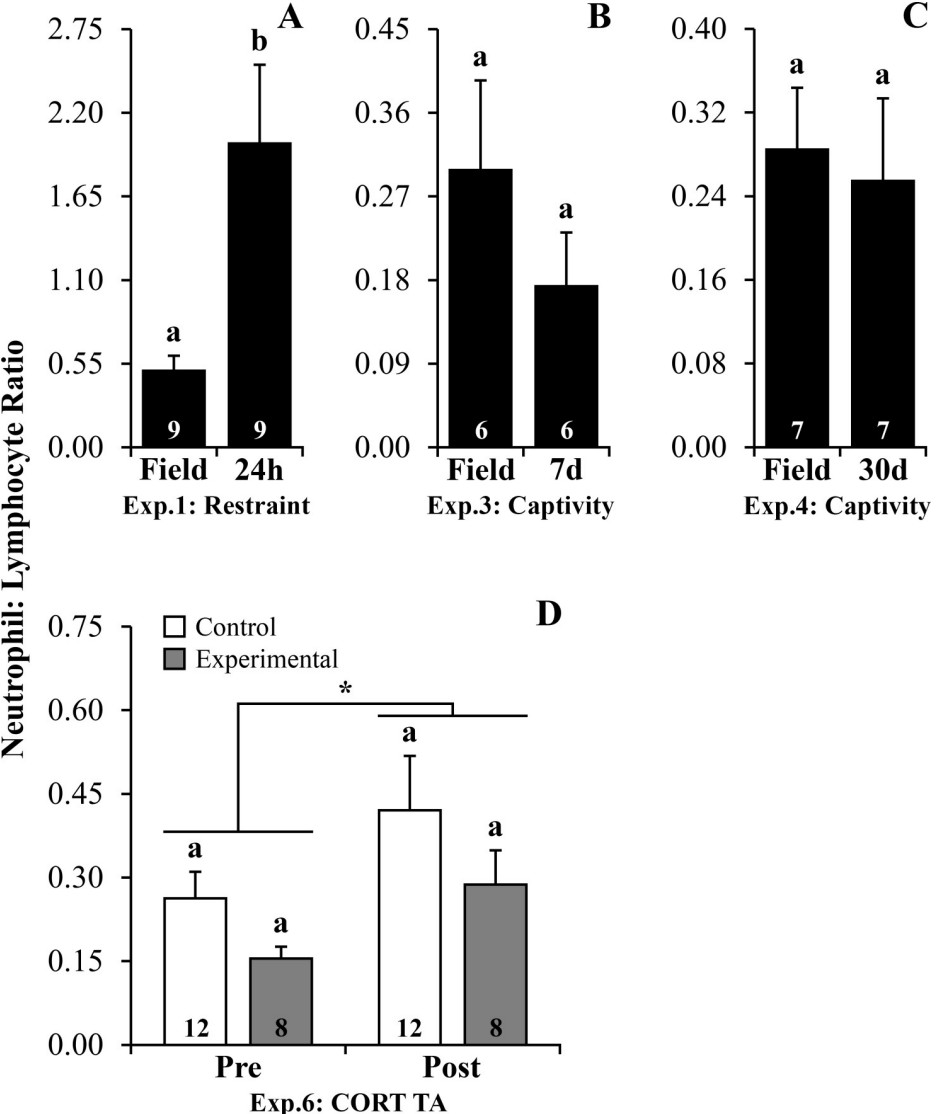

**Fig 2. Neutrophil Lymphocyte ratio following stressors and corticosterone transdermal application in *R. ornata* toads.** (A) Experiment 1: Restraint–field *vs*. 24h. (B) Experiment 3: Captivity–field *vs*. 7 days. (C) Experiment 4: Captivity–field *vs*. 30 days. (D) Experiment 6: Corticosterone transdermal application. Bars represent mean ± standard error. The N is indicated inside each bar. Letters above the bars represent statistical differences for Bonferroni adjustments in ANOVAs or Student t-test, with different letters representing statistical difference within groups with $P \leq 0.05$. Asterisk represents statistical differences between pre and post-treatment, independently of the treatment (Control and experimental groups). Abbreviation: **Exp.:** Experiment; **CORT TA:** Corticosterone transdermal application. Legend: **Control:** control and placebo group; **Experimental:** corticosterone group.

As expected, for males showing low CORT at 0h, restraint increased CORT plasma levels, as previously described for anurans including *R. ornata* [11,13,22,23,30]. Additionally, our results showed maximum CORT values 1h of after restraint compared to 24h, which might indicate negative feedback of the hypothalamic-hypophyseal-interrenal axis [31]. Meanwhile, for the individuals in the field displaying high baseline CORT, we observed a tendency to decrease CORT plasma levels following 24h restraint. These toads were already displaying high CORT in the field probably due to reproductive behavior (calling activity) [27], precluding the assessment of increased CORT values 24h post-restraint [11]. Moreover, the trend to

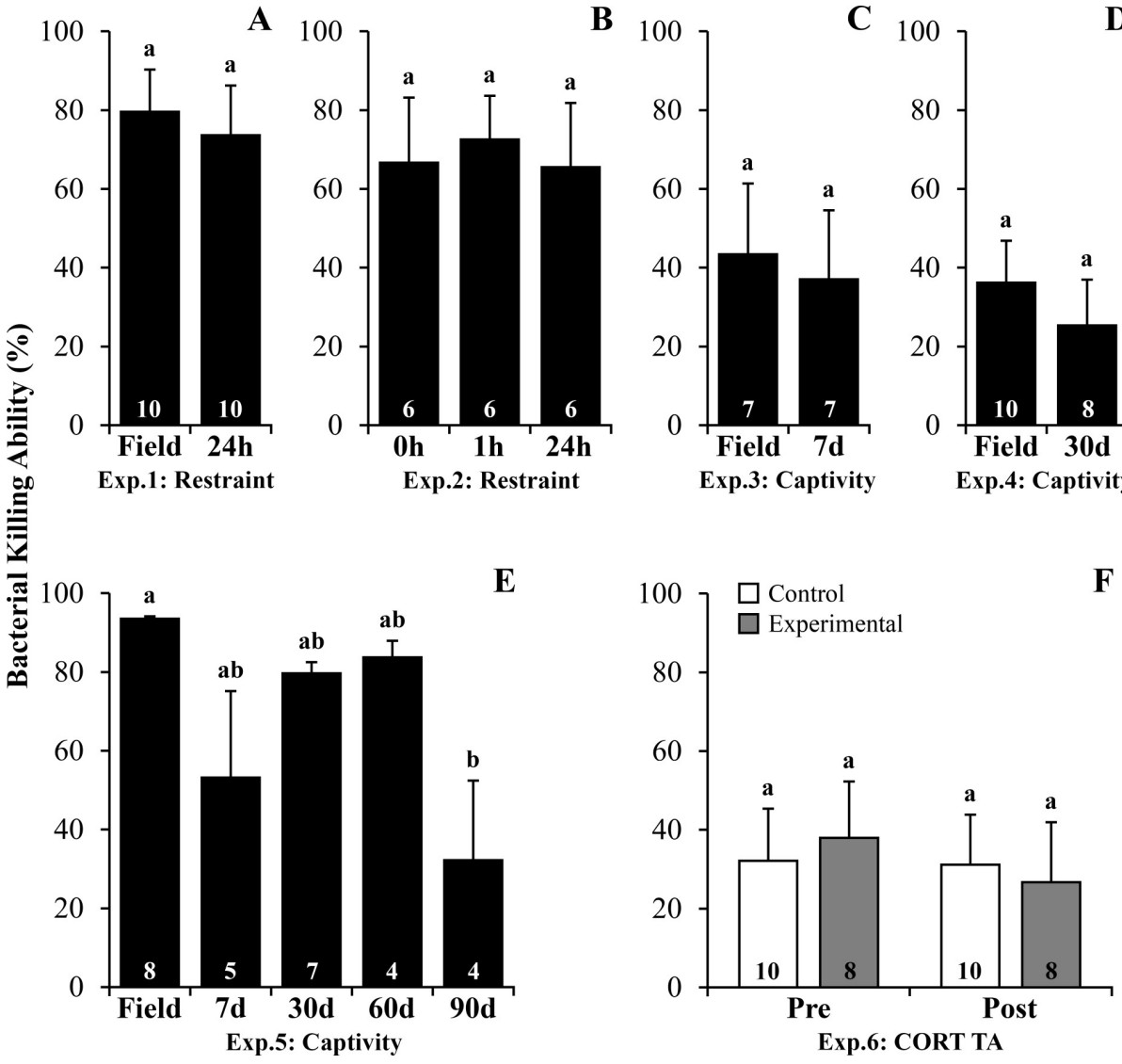

**Fig 3. Bacterial killing ability following stressors and corticosterone transdermal application in *R. ornata* toads.** (A) Experiment 1: Restraint–field *vs*. 24h. (B) Experiment 2: Restraint– 0h *vs*. 1h *vs*. 24h. (C) Experiment 3: Captivity–field *vs*. 7 days. (D) Experiment 4: Captivity–field *vs*. 30 days. (E) Experiment 5: Captivity–field *vs*. 7 *vs*. 30 *vs*. 60 *vs*. 90 days. (F) Experiment 6: Corticosterone transdermal application. Bars represent mean ± standard error. The N is indicated inside each bar. Letters above the bars represent statistical differences for Bonferroni adjustments in ANOVAs or Student t-test, with different letters representing statistical difference within groups with $P \leq 0.05$. Abbreviation: **Exp.:** Experiment; **CORT TA:** Corticosterone transdermal application. Legend: **Control:** control and placebo group; **Experimental:** corticosterone group.

decreased CORT 24h post-restraint in these toads also corroborates the observation that the peak of GC release in response to restraint occurs before 24h of exposure to this stressor. *Rhinella ornata* toads that were engaged in calling activity and displayed high CORT in the field also showed decreased CORT plasma levels in response to captivity maintenance, which could be associated with the interruption of vocal behavior [27,32]. In addition, neither short nor long-term captivity elicited NLR changes in these toads. The absence of response in NLR along with low CORT plasma levels, suggest that these *R. ornata* males were less affected by captivity than males from other species in the same genus exposed to captivity [21,27,28]. We also described changes in CORT plasma levels following CORT TA, with increased CORT in the

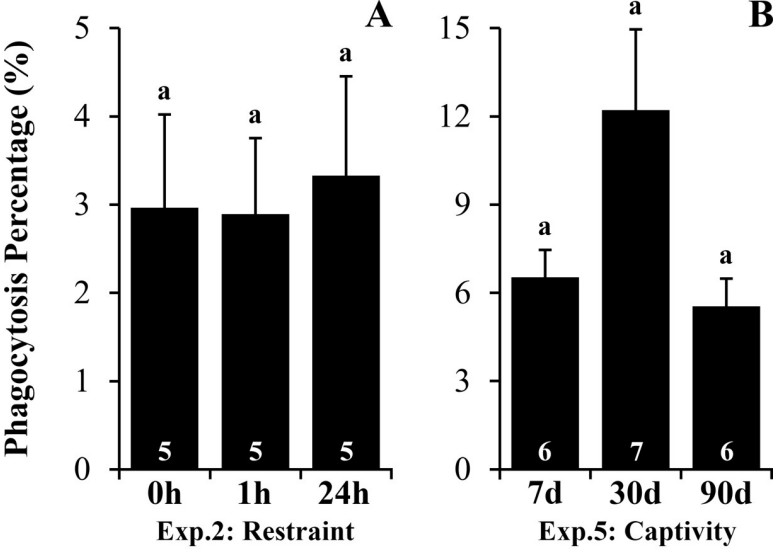

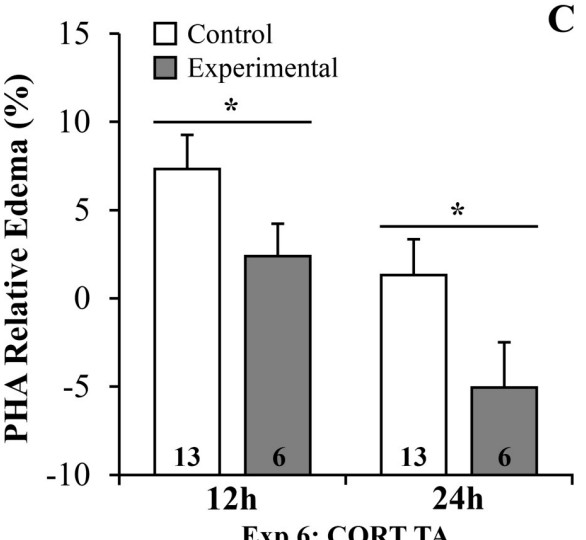

**Fig 4. Phagocytosis activity and inflammatory response following stressors and corticosterone transdermal application in *R. ornata* toads.** (A) Experiment 2: Restraint– 0h *vs*. 1h *vs*. 24h. (B) Experiment 5: Captivity– 7 *vs*. 30 *vs*. 90 days. Phytohemagglutinin relative edema. (C) after 12 and 24h post PHA injection at the end of Experiment 6: Corticosterone transdermal application. Bars represent mean ± standard error. The N is indicated inside each bar. Letters above the bars represent statistical differences for Bonferroni adjustments in ANOVAs, with different letters representing statistical difference within groups with $P \leq 0.05$. The asterisks represent ANCOVA statistical differences ($P \leq 0.05$) between treatment groups (control *vs*. experimental) at each specific time (12 or 24h). Abbreviation: **Exp.:** Experiment; **CORT TA:** Corticosterone transdermal application; **PHA:** Phytohemagglutinin. Legend: **Control:** control and placebo group; **Experimental:** corticosterone group.

experimental group 14 times higher than the control group after 20 consecutive days of CORT TA. Although CORT plasma levels after CORT TA were extremely high in the experimental group (348 ± 54ng/ml, mean ± standard error), high CORT levels (282 ± 52ng/ml, mean ± standard error) were reported for this same species following severe dehydration [13], evidencing these animals are physiologically able to reach such high circulating CORT levels.

**Table 1. Stress and corticosterone transdermal application-related corticosterone plasma levels and immune modulation of *R. ornata* toads.**

| Treatment/Variable | CORT | NLR | BKA | CELULAR |
|---|---|---|---|---|
| Restraint (1, 24h) | ↑ (Ø), ↓ (+) | (+) | (Ø) | PP (Ø) |
| Short-term captivity (7 days) | ↓ (Ø), ↑ (—) | (Ø) | (Ø) | |
| Mid-term captivity (30 days) | ↑ (—) | (Ø) | (Ø) | PP (+) |
| Long-term captivity (≥ 60 days) | ↑ (—) | (Ø) | (—) | |
| CORT transdermal application (20 days) | ↓ (+) | (+) | (|) | PHA (—) |

Abbreviation as follow: **CORT:** corticosterone; **NLR:** neutrophil: lymphocyte ratio; **BKA:** bacterial killing ability; **PHA:** phytohemagglutinin relative edema; **PP:** phagocytosis percentage. Symbols: **(+):** increased; **(—):** decreased; **(Ø):** not significantly altered; **(↓):** low corticosterone in the field or time 0; **(↑):** high corticosterone in the field or time 0.

Increased NLR was observed after restraint challenge (24h) and submission to the CORT TA experiment (both control and experimental groups) for 20 days. Additionally, in CORT TA experiment, we did not find differences between control and CORT group, which might indicate that daily manipulation itself is a stressor for *R. ornata*. Increased NLR following restraint challenge and post-CORT TA were reported for *Rhinella* toads, including *R. ornata* [11,12]. Increased NLR was also described for salamanders 50h post-capture following both ACTH and saline injection, even with a concomitant decrease in CORT [33]. Accordingly, the increased NLR observed in *R. ornata* following restraint challenge and 20 days post-TA was not associated with changes in CORT, even considering that TA-induced increase in CORT plasma levels was so prominent. Besides the prominent increase in CORT plasma levels resulting from CORT TA, toads from the control group also showed a tendency to increase CORT throughout the experiment. This tendency might indicate that daily manipulation (box opening and water removal and return) could represent a stressor for all groups, resulting in increased NLR for both control and CORT TA groups. In general, although we observed both increased CORT plasma levels and NLR in response to the set of different stressors applied in the present study, these response variables could not be easily associated, as already observed in other contexts and study models [8].

Our results demonstrate preserved BKA in almost all treatments (restraint, short and mid-term captivity and TA) except after long-term captivity, given that a reduction in values was observed in the 90th day of captivity maintenance. Plasma BKA is an innate, non-cellular immune response mediated through the complement system and lysozymes, both of which promote lysis of invading microorganisms [34,35]. Studies conducted in birds, reptiles and mammals have demonstrated both acute and/or prolonged regulation of BKA-associated proteins following stressors [36–39]. In amphibians, impaired BKA has been described following acute [23] and long-term [21,27,28] stressors and GCs exogenous treatment [21]. However, temporal patterns of BKA stress-associated responses are diverse. Decreased BKA following long-term captivity was transient for *R. schneideri*, but sustained for *R. icterica* [27,28]. The absence of changes in response to most of the short and mid-term stressors and CORT treatment, along with decreased values only after long-term captivity maintenance, suggest that BKA presents robust immune mediators in *R. ornata* toads.

Regarding immune cellular activity, we did not observe changes in blood PP 1h and 24h after restraint in *R. ornata*. In contrast, increased phagocytosis 13h after restraint was described in another *Rhinella* species [23]. Thus, the lack of response in PP 1 and 24h post restraint could indicate these time points are ideal for ascertaining how restraint influences any blood phagocytosis alterations in *R. ornata* toads [1,2]. Meanwhile, phagocytosis of peritoneal cells was affected by mid-term captivity, with increased mean values for PP on the 30th

day. Up-regulation of immune responses following stressors, including phagocytic activity, is often observed in several vertebrates, but in a context of acute stress [1,2,14,40,41]. Results of phagocytic stress-driven changes are unclear as studies have presented conflicting results [27,28]. Meanwhile, increased phagocytosis was previously observed in response to 13h restraint [23] and after a month following CORT treatments, such as CORT TA [12], capsule implant and immersion in CORT-treated water [4]. Our study showed maximum values for PP on the 30th day compared to the 7th and 90th day in captivity. Moreover, mean CORT values were maximum on the 30th day compared to the other days in captivity, which could be associated with the increased phagocytic activity following mid-term captivity. Specifically Falso et al. (2015) [4], found that CORT treatment was associated with increased phagocytosis and increased neutrophil concentration in the blood (a parameter not measured in our captivity experiment–field to 90 days), along with high CORT levels [4]. These results indicate a complex relationship between stress and immunity, and the importance in accounting for time-related changes of distinct branches of immune system in response to a stress challenge in anurans.

The inflammatory response, measured as PHA relative edema, was affected by the CORT TA at both 12h and 24h in *R. ornata* toads. The inflammatory and cell-mediated immune response is strongly associated with stress intensity (high CORT and catecholamine levels) and duration (reviewed in [2]). While an acute increase in CORT stimulates the inflammatory response, persistent GCs elevations suppress it [2,15,17]. The inflammatory response can be suppressed at multiple levels, including the inhibition of leukocyte recruitment or reduced leukocyte pro-inflammatory cytokine expression, among others [2,16]. In amphibians, chronic treatment with high doses of CORT delays cutaneous wound healing in salamanders [42] and frogs (Madelaire et al., unpublished data). Therefore, the acute daily increase in CORT plasma levels, by CORT TA, could have promoted an anti-inflammatory effect in *R. ornata* toads, a common effect following chronical or several acute elevations of GCs in mammals [2].

## Conclusions

In summary, different time-related trajectories for CORT plasma levels and immune parameters were described in *R. ornata* toads following stressors and CORT TA. Since CORT levels were sustained (for those animals with low levels in the field) or decreased (for those individuals displaying high levels in the field) in response to short and mid-term captivity, along with no changes in BKA and NLR, it appears that captivity for a 30 days does not represent a stressor for this species. However, mid and long-term captivity increased phagocytosis activity of peritoneal cells in the 30th and decreased BKA in the 90th day, respectively, indicating captivity promotes immune changes, with captivity maintenance for 90 days potentially representing a low-intensity stress condition. In the meantime, CORT changes along with increased NLR indicate a stress response following restraint challenge. Moreover, increased CORT associated with a reduction in PHA swelling response after 20 consecutive days of CORT TA, indicate a chronic stress situation for *R. ornata*. Together, these results demonstrate immuno-enhancing effects following restraint, short and mid-term stressors, while long-term stressors and CORT administration promoted immunosuppression. Additionally, our results indicate cellular immunity are more responsive to stressors and high CORT levels induced by CORT TA than non-cellular immune response in *R. ornata* toads. Therefore, we encourage researchers working within the ecoimmunology area, at the field and under laboratory conditions, to conduct studies that include a number of different stress protocols and sampling at multiple time-points, to better understand patterns of stress-related immunomodulation in vertebrates.

## Materials and methods

### Study sites and animals

Adult males of *R. ornata* were collected at the Biosciences Institute of University of Sao Paulo, Sao Paulo/Brazil (23˚33'51.3"S, 46˚43'48.5"W) in August 2012 (*N* = 27) and 2018 (*N* = 7), Botanic Garden of Sao Paulo, Sao Paulo/Brazil (23˚38'19.8"S, 46˚37'18.5"W) in August 2016 (*N* = 10), and in Botucatu (22˚53'21.1"S 48˚28'20.1"W) Sao Paulo/Brazil in August 2014 (*N* = 24) and August 2016 (*N* = 42). Toads were collected under license from the Instituto Chico Mendes de Conservação da Biodiversidade (ICMBio n˚ 29896). The Ethic committee of the Biosciences Institute of University of Sao Paulo approved all procedures for this study (CEUA n˚ 142/2011; 054/2013; 249/2016).

### Field procedures

Toads were visually located and hand captured. Blood samples were collected in August 2012 (*N* = 7), August 2014 (*N* = 20) and August 2016 (*N* = 21), for baseline (field) measurements (CORT plasma levels, NLR and BKA). All toads were weighed (to the nearest 0.01g) and measured (to the nearest 0.01mm) after capture. Animals were released at their respective collection point following restraint challenge and fieldwork, or were kept in individual plastic containers and then transported to the laboratory. Detailed collection and experimental information are in Table 2.

### Restraint challenge (Experiment 1: Field *vs.* 24h and Experiment 2: 0h *vs.* 1h *vs.* 24h)

Immediately after blood collection in the field for baseline measurements (August 2014), the same toads (*N* = 20) were exposed to restraint challenge (Experiment 1), by placing individuals into wet cloth bags and then inside individual plastic bins (29 x 18 x 15 cm) for 24h according to [11,21]. All bins had lids with holes to allow air circulation. After 24h, all toads were bled again to reassess the same physiological variables measured at baseline (CORT, NLR and BKA). Upon termination of this experimental protocol, toads were weighed (body mass, 0.01 g), and returned to their collection point at night.

*Rhinella ornata* toads (August 2018) kept in captivity for 7 days, under the same conditions as toads in experiment 4 and 5, were bled (0h), then immediately placed into wet cloth bags and then inside individual plastic bins (29 x 18 x 15 cm) for 24h according to [11,21]. These same animals were bled again 1 and 24h after the restraint protocol (Experiment 2) in order to compare physiological variables measured at the three-sampling time (CORT, NLR, BKA and blood PP). Upon termination of this experimental protocol, toads were weighed (body mass, 0.01 g), and returned to their captivity conditions for further studies.

### Short-term captivity (Experiment 3: Field *vs.* 7 days) and Corticosterone transdermal application (Experiment 6: Control *vs.* placebo *vs.* corticosterone)

Captive toads collected in August 2012 (*N* = 24) were kept individually in plastic bins (43.0 x 28.5 x 26.5 cm) with free access to water and lids with holes to allow air circulation. Each toad was fed cockroaches once per week and provided ad libitum access to water. The room was kept on 11/13 LD photoperiod (lights on at 7:40 am and off at 6:40 pm) at a temperature of 21 ± 2˚C. After 7 days, a blood sample was collected from each individual, to evaluate CORT, NLR and BKA (Experiment 3). Toads were also weighed by the initial and last day of the experiment.

**Table 2. Detailed information regarding year of collection and experimental procedures, statistical comparisons and studied parameters of *R. ornata* toads.**

| Year | Collection Site | Experiment | Comparison | Parameters |
|---|---|---|---|---|
| | Botucatu | Experiment 1 | Field *vs*. 24h | CORT |
| 2014 | (22°53'21.1"S 48°28'20.1"W) | Restraint | post restraint | NLR |
| | | (24h) | (Repeated measure) | BKA |
| | University of Sao Paulo | Experiment 2 | 0h *vs*. 1h *vs*. 24h | CORT |
| 2018 | (23°33'51.3"S, 46°43'48.5"W) | Restraint | post restraint | BKA |
| | | (1 and 24h) | (Repeated measure) | PP |
| | University of Sao Paulo | Experiment 3 | Field *vs*.7 days | CORT |
| 2012 | (23°33'51.3"S, 46°43'48.5"W) | Short-term captivity | (Repeated measure) | NLR |
| | | (7days) | | BKA |
| | Botanic Garden of Sao Paulo | Experiment 4 | Field *vs*. 30 days | CORT |
| 2016 | (23°38'19.8"S, 46°37'18.5"W) | Mid-term captivity | (Repeated measure) | NLR |
| | | (30 days) | | BKA |
| | Botucatu | Experiment 5 | Field *vs*. 7 *vs*. 30. | CORT |
| 2016 | (22°53'21.1"S 48°28'20.1"W) | Long-term captivity | *vs*. 60 *vs*. 90 days | BKA |
| | | (7, 30, 60, 90 days) | (univariate) | PP |
| | University of Sao Paulo | Experiment 6 | Pre *vs*. Post treatment | CORT |
| 2012 | (23°33'51.3"S, 46°43'48.5"W) | CORT TA | (control, placebo, | NLR |
| | | (20 days) | experimental) | BKA |
| | | | (Mixed) | PHA |

Abbreviation as follow: **CORT TA:** corticosterone transdermal application; **CORT:** corticosterone plasma levels; **NLR:** neutrophil: lymphocyte ratio; **BKA:** bacterial killing ability; **PHA:** phytohemagglutinin relative edema; **PP:** phagocytosis percentage.

To perform the CORT TA (Experiment 6), toads were divided in three groups according to [21]: control (no treatment), placebo (5ul sesame oil application on the back of the animals) and experimental (5ul CORT [Sigma C27840] + sesame oil application– 3ug CORT /1ul sesame oil). Following [21] protocol, all the animals received the same amount of CORT solution, since no relationship was found between body mass and CORT in these toads in field conditions ($\rho$ = -0.594; *P* = 0.160). Water was removed from each box to ensure complete absorption of the treatment (vehicle or CORT solution) for all groups before daily application and was returned 3h after treatment. CORT TA occurred for 20 consecutive days. On the last day, a new blood sample was collected from each animal, 1h after the TA, to evaluate the same variables as in a week in captivity (Experiment 3: CORT, NLR and BKA). In order to assess an inflammatory response, one day after the end of the CORT TA, all individuals were exposed to the PHA immune challenge.

## Mid-term captivity (Experiment 4: Field *vs*. 30 days)

Captive toads collected in August 2016 were kept individually in plastic bins (43.0 x 28.5 x 26.5 cm) with leaf litter as a substrate and provided ad libitum access to water. The lids of the bins had holes to allow air circulation. Each toad was fed with cockroaches once per week. The room was kept on 11/13 LD photoperiod (lights on at 7:40 am and off at 6:40 pm) and temperature of 21 ± 2°C.

Toads from Botanical Garden of Sao Paulo (*N* = 10) were bled again only after 30 days in captivity (Experiment 4), in order to evaluate the effects of 30 days in captivity on plasma CORT levels, BKA and NLR.

### Long-term captivity (Experiment 5: 0 *vs*. 7 *vs*. 30 *vs*. 60 *vs*. 90 days)

Toads from Botucatu ($N$ = 29) were kept under the same conditions described for toads in experiment 4 and divided in four groups to be sampled after 7, 30, 60, and 90 days under captivity (Experiment 5), in order to evaluate the effects of time in captivity on plasma CORT levels, BKA and peritoneal PP according [27,28]. After bleeding on each specific day (7, 30, 60 and 90), animals were euthanized with an intraperitoneal injection (75mg/kg) of sodium thiopental (Thiopenthax®) solution (25mg/ml).

### Blood collection

Blood samples from each individual were collected (~ 140 ul) by cardiac puncture using a heparinized 1 ml syringe and 26Gx1/2" needle within 3 minutes of capture. Blood samples were collected, kept on ice (< 2 hours) until blood smears could be performed (for determine the NLR), and then centrifuged (4 min at 604 g) to isolate the plasma. Plasma samples were stored at -80˚C freezer until analysis.

### Neutrophil: Lymphocyte ratio (NLR)

A drop of blood was used to perform each blood smear. The slide was stained with Giemsa solution (10%) and observed under an optical microscope (100X objective, using oil immersion—Nikon E200, 104c). One hundred leukocytes were counted on each slide, and classified based on morphology as neutrophils, lymphocytes, eosinophils, basophils, and monocytes [43]. The NLR was calculated as the number of neutrophils divided by the number of lymphocytes on each slide.

### Bacterial killing ability (BKA)

The BKA assay was conducted according [44]. Briefly, plasma samples diluted in amphibian Ringer's solution (10 ul plasma: 190 ul Ringer) were mixed with 10 ul of *Escherichia coli* working solution (~ $10^4$ microorganisms, *E. coli*; ATCC 8739). Positive controls consisted of 10 ul of *E. coli* working solution in 200 ul of Ringer's solution, and negative control contained 210 ul of Ringer's solution. All samples and controls were incubated by 60 min at 37˚C (optimal temperature for bacterial growth). Thereafter, 500 ul of tryptic soy broth were added to each sample, vortexed and transferred (in duplicates) to a 96 wells microplate (300 ul per well), which was incubated at 37ºC for 2h. Following that, the optical density of the samples was measured hourly in a plate spectrophotometer (wavelength 600 nm). The BKA was evaluated at the beginning of the bacterial exponential growth phase and calculated according to the formula: *1 - (optical density of sample / optical density of positive control)*, which represents the proportion of killed microorganisms in the samples compared to the positive control.

### Phagocytosis

In order to assess the cellular immune measure, we performed a phagocytosis assay with blood (experiment 2) or peritoneal (experiment 5) leukocytes. Phagocytic activity was evaluated as previously described by [28]. The phagocytosis assay was performed by adding 100 ul of zymosan (Sigma Z4250) labeled with green fluorescence (CFSE Sigma 21888F) suspension (1 x $10^6$ particles) in 1000 ul of APBS containing 2 x $10^5$ cells (macrophages/neutrophils). Samples were incubated for 60 min, under agitation at 25˚C, and then washed with 2mL of cold (4˚C) 6mM of ethylenediamine tetra acetic acid (EDTA) solution, in order to stop phagocytosis and to wash off the excess of free zymosan particles. After centrifugation (4˚C, 259g, 7min), pellets were ressuspended in 200 ul of cold (4˚C) paraformaldehyde (1%) for cell fixation. After 1

hour, 500ul of APBS were added, samples were centrifuged (4˚C, 259g, 7min) and pellets were ressuspended in 100ul of APBS for flow cytometry.

Cells were analyzed on an image flow cytometer (AMNIS Flowsight imaging flow cytometer Merck-Millipore, German) interfaced with a DELL computer with 10,000 events collected using the 488nm laser at a 20x magnification, through INSIRE software. Data analysis was performed using IDEAS analysis software (EMD Millipore) version 6.1 for windows. After gating on side scatter *vs.* brightfield plot, phagocytosis of peritoneal cells was measured by phagocytosis percentage (PP), which represents the percentage of cells that engulfed at least one zymosan particle (the percentage of green positive cells).

## Phytohemagglutinin (PHA) swelling assay

The PHA challenge was performed in order to assess the inflammatory response 24h after the TA (Exp. 6) in *R. ornata*, according to [21,45]. Specifically, the hind fleshy base of the right foot was injected with 10ul of PHA (20 mg PHA [Sigma L8754]/ml sterile saline solution) using a 10 ul glass syringe and 30Gx1/2″ needle. The hind fleshy base of the left foot of the same animal was injected with 10 ul of sterile saline solution, as a control. The thickness of each injected hind fleshy base of each animal was measured with a thickness gauge (Digimess, accuracy 0.01 mm) before, 12 and 24h after PHA and saline injections. Measurements consisted of three successive thickness measurements. At each time point, three successive measurements of thickness were taken, with the mean of these values was used in analysis. The swelling in response to PHA and saline was calculated from the proportional change in foot thickness before, 12 and 24h after injection. The relative swelling was considered as the thickness in 12 or 24 hours divided by initial thickness multiplied by 100. The PHA relative edema was calculated as the PHA relative swelling minus the saline relative swelling.

At the end of the 24h swelling measurement, individuals were euthanized with an intraperitoneal injection (75mg/kg) of sodium thiopental (Thiopenthax®) solution (25mg/ml).

## Corticosterone assays

Steroid hormones were initially extracted with ether according to [11,21]. CORT were determined using EIA kits (CORT number 501320; Cayman Chemical), according to the manufacturer's instructions and previous studies conducted with *Rhinella* toads, including *R. ornata* species [11]. The mean values for intra and inter-assay variation were 6.83% and 4.77% respectively. Sensitivity of the assays was 32 pg/ml.

## Statistical analyses

All statistical analyses were performed in IBM SPSS Statistics 22. Normality of all data were assessed via the Shapiro-Wilk normality test. With the exception of BKA in restraint challenge and captivity treatments, all variables were normal.

Sets of ANCOVAs for independent or repeated measures (as applicable for each set of data, Table 2) were used to investigate the effect of each experiment (1–6: Restraint challenge, Captivity and TA) on studied variables. CORT, NLR, BKA, PHA relative edema and PP were used as dependent variables, body mass and snout-vent length as co-variables and each treatment and/or time as a factor. Unstandardized residuals of a linear regression of body mass as a function of snout-vent length were used as body index and also included as co-variable in the ANCOVAs.

For the dependent variables not significantly affected by body mass, snout-vent length and body index a set of ANOVAs was then performed with treatment (Experiment 2: Restraint hour [0, 1, 24h]; Experiment 5: captivity days [field, 7, 30, 60, 90]; or Experiment 6: TA group

[control, placebo, experimental]) as factor. The ANOVAs were followed by tests for mean multiple comparisons with Bonferroni adjustments. Paired comparisons with Student-t test were performed for the dependent variables not significantly affected by body mass in the experiments with only two comparisons (Experiment 1: Restraint [baseline *vs*. 24h post restraint]; Experiment 3 and 4: captivity [field *vs*. 7 days and field *vs*. 30 days]). Since BKA showed absence of normality, we used the Wilcoxon Signed Ranks test, which is a non-parametric test.

Full raw data are available in the S1 File.

## Supporting information

**S1 Table. Corticosterone plasma levels student-t test after stressors in *Rhinella ornata* toads.** Effect of restraint challenge (Exp. 1) and captivity (Exp. 3 and 4) on corticosterone plasma levels tested through repeated measures student-t tests on *R. ornata*.
(DOCX)

**S2 Table. Corticosterone plasma levels analysis of variance after stressors and corticosterone transdermal application in *R. ornata* toads.** Effect of restraint challenge (Exp. 2), captivity duration (Exp. 5) and corticosterone transdermal application (Exp. 6) on plasma corticosterone levels of *R. ornata* tested through a set of ANOVAs, with plasma corticosterone levels as dependent variable, hour (0, 1, 24h), captivity duration (field, 7, 30, 60 and 90 days), group (control and corticosterone), and time (pre-experiment and post-experiment) and as factors.
(DOCX)

**S3 Table. Corticosterone plasma levels analysis of variance after corticosterone transdermal application in *R. ornata* toads.** Effect of corticosterone transdermal application (Exp. 6) on plasma corticosterone levels of *R. ornata* tested through a set of mixed ANOVAs, with plasma corticosterone levels as dependent variable and group (control, placebo and corticosterone) and time (pre-experiment and post-experiment) as factors.
(DOCX)

**S4 Table. Corticosterone plasma levels analysis of variance after stressors and corticosterone transdermal application in *R. ornata* toads with Bonferroni comparisons.** Bonferroni comparisons of mixed ANOVAs for corticosterone transdermal application (Exp. 6) on plasma corticosterone levels of *R. ornata*, with plasma corticosterone levels as dependent variable and group (control, placebo and corticosterone) and time (pre-experiment and post-experiment) as factors.
(DOCX)

**S5 Table. Neutrophil: Lymphocyte ratio student-t test after stressors in *Rhinella ornata* toads.** Effect of restraint challenge (Exp. 1) and captivity (Exp. 3 and 4) on neutrophil: lymphocyte ratio tested through repeated measures student-t tests on *R. ornata*.
(DOCX)

**S6 Table. Neutrophil: Lymphocyte ratio analysis of variance after corticosterone transdermal application in *R. ornata* toads.** Effect of corticosterone transdermal application (Exp. 6) on neutrophil: lymphocyte ratio of *R. ornata* tested through a set of mixed ANOVAs, with neutrophil lymphocyte ratio as dependent variable and group (control and corticosterone) and time (pre-experiment and post-experiment) as factors.
(DOCX)

**S7 Table. Plasma bacterial killing ability student-t test after stressors in *Rhinella ornata* toads.** Effect of restraint challenge (Exp. 1) and captivity (Exp. 3 and 4) on bacterial killing ability tested through Wilcoxon Signed Ranks test on *R. ornata*.
(DOCX)

**S8 Table. Plasma bacterial killing ability analysis of variance after stressors and corticosterone transdermal application in *R. ornata* toads.** Effect of restraint challenge (Exp. 2), captivity duration (Exp. 5) and corticosterone transdermal application (Exp. 6) on bacterial killing ability of *R. ornata* tested through a set of ANOVAs, with bacterial killing ability as dependent variable, hour (0, 1, 24h), captivity duration (field, 7, 30, 60 and 90 days), group (control and corticosterone), and time (pre-experiment and post-experiment) as factors.
(DOCX)

**S9 Table. Phagocytosis percentage analysis of variance after corticosterone transdermal application in *R. ornata* toads.** Effect of Restraint Challenge (Exp. 2) and captivity duration (Exp. 5) on phagocytosis of *R. ornata* tested through a set of univariate ANOVAs, with phagocytosis percentage and phagocytosis efficiency as dependent variables and time (0, 1 and 24h) and captivity duration (field, 7, 30, and 90 days) as factors.
(DOCX)

**S10 Table. Phytohemaglutinin relative edema analysis of variance after corticosterone transdermal application in *R. ornata* toads.** Effect of corticosterone transdermal application (Exp. 6) on phytohemagglutinin relative edema of *R. ornata* tested through a set of mixed ANCOVAs, with phytohemagglutinin relative edema as dependent variable, body mass as co-variable and time (12 and 24 hours post injection) and group (control and corticosterone) as factors.
(DOCX)

**S1 File. Data availability file.** File containing full raw data of the six performed experiments in this study.
(DOCX)

## Acknowledgments

We thank Antônio Pereira de Sousa for allowing us to collect on their property (Chácara Santo Antônio, Botucatu/SP), the lab technicians Eduardo Braga and Wagner Alberto for their help, and the use of the AMNIS Flowsight imaging flow cytometer from the Laboratory of Chrono-pharmacology, Bioscience Institute, University of Sao Paulo, Brazil. We thank Ph.D. John W. Finger (Department of Biological Sciences, Auburn University, Auburn, AL/USA) for assisting with the English grammar review.

## Author Contributions

**Conceptualization:** Stefanny Christie Monteiro Titon, Fernando Ribeiro Gomes, Vania Regina Assis.

**Data curation:** Stefanny Christie Monteiro Titon, Braz Titon Jr, Vania Regina Assis.

**Formal analysis:** Stefanny Christie Monteiro Titon, Braz Titon Jr.

**Funding acquisition:** Fernando Ribeiro Gomes, Vania Regina Assis.

**Investigation:** Stefanny Christie Monteiro Titon.

**Methodology:** Stefanny Christie Monteiro Titon, Braz Titon Jr, Adriana Maria Giorgi Barsotti, Vania Regina Assis.

**Project administration:** Stefanny Christie Monteiro Titon.

**Resources:** Adriana Maria Giorgi Barsotti.

**Supervision:** Fernando Ribeiro Gomes.

**Writing – original draft:** Stefanny Christie Monteiro Titon, Fernando Ribeiro Gomes, Vania Regina Assis.

**Writing – review & editing:** Braz Titon Jr, Adriana Maria Giorgi Barsotti.

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
