## [Decision Letter · Decision Letter 0]

8 Aug 2019

PONE-D-19-16526

Time-related immunomodulation by stressors and corticosterone transdermal application in toads

PLOS ONE

Dear Dr. Monteiro Titon,

Thank you for submitting your manuscript to PLOS ONE. After careful consideration, we feel that it has merit but does not fully meet PLOS ONE’s publication criteria as it currently stands. Therefore, we invite you to submit a revised version of the manuscript that addresses the points raised during the review process.

ACADEMIC EDITOR: This is a sound and very well-written manuscript on the effects of different stressors and of corticosterone on immune responses in toads. Based on the reviewers' comments below, the manuscript will require minor revisions, especially in relation to the times and locations of sampling, the consideration of body size in some of the assessments, and the discussion of the effects of corticosterone transdermal application on immune responses. 

We would appreciate receiving your revised manuscript by Sep 22 2019 11:59PM. To enhance the reproducibility of your results, we recommend that if applicable you deposit your laboratory protocols in protocols.io, where a protocol can be assigned its own identifier (DOI) such that it can be cited independently in the future. For instructions see: http://journals.plos.org/plosone/s/submission-guidelines#loc-laboratory-protocols

We look forward to receiving your revised manuscript.

Kind regards,

Marie-Claude Audet

Academic Editor

PLOS ONE

Journal Requirements:

Reviewers' comments:

Reviewer's Responses to Questions

**Comments to the Author**

1. Is the manuscript technically sound, and do the data support the conclusions?

Reviewer #1: Yes

Reviewer #2: Yes

2. Has the statistical analysis been performed appropriately and rigorously? 

Reviewer #1: Yes

Reviewer #2: Yes

3. Have the authors made all data underlying the findings in their manuscript fully available?

Reviewer #1: Yes

Reviewer #2: Yes

4. Is the manuscript presented in an intelligible fashion and written in standard English?

Reviewer #1: Yes

Reviewer #2: Yes

5. Review Comments to the Author

Reviewer #1: 307: Please explain why the sampling dates (years) are spreaded so much ?

308: Why sampling was done across different locations ?

313: I can see ethics approval for 2013 and 2016 only. Did this ethics approval cover the 2018 sampling as well ?

Reviewer #2: This study examines temporal changes in immune responses and plasma corticosterone levels following restraint, captivity, and corticosterone administration in toads. Given the multiple immune responses that were measured (neutrophil: lymphocyte ratios, plasma bacterial killing ability, phagocytic activity of peritoneal cells, and phytohemagglutinin swelling response) and the various time frames in which they were measured, this study represents one of the most thorough studies on the topic in amphibians. Overall, the results indicate that short term stress enhances immune function while long-term stress suppresses immune function. The paper is well written and the introduction and results are clear and well thought out. I only have a few comments/questions that are specified below:

Methods

Line 316. Correct “toads were located by visually and captured by hand”.

The relationship between body mass and CORT was examined but was body size measured? I’m wondering how the authors accounted for differences in mass that were attributable to size. For example, there was no relationship between body mass and CORT (Line 359 and elsewhere) but would it be better to examine body condition estimates that factor in size differences?

Line 451. How many CORT assays were run? Were the assays validated for this species?

Results

Line 99. Body mass was not manipulated so it would be better to state that there was no relationship between body mass and the measured variables. What was the direction of the relationship with PHA?

Line 102. How were toads with high and low initial CORT levels delineated (i.e., how did you determine the cutoff point)?

Line 111. Rephrase to state that CORT levels were 14 times higher in TA-treated toads compared to controls.

CORT levels increased dramatically in TA-treated toads (Fig 1E). The authors mention the upper limits of CORT in this species following severe dehydration (line 221), but can the authors comment about the natural range of CORT levels and whether the effects on immune responses (especially PHA measures shown in Fig. 4C) actually occur within the natural range of CORT levels (i.e., CORT levels in the TA treatment group is still quite a bit higher than the upper limits reported)?

Line 125. Of the 9 individuals wherein NLR was measured after constraint (Fig. 2A), do these represent individuals that showed an increase or decrease in CORT levels during constraint (see Fig. 1 A and B)?

6. PLOS authors have the option to publish the peer review history of their article (what does this mean?). If published, this will include your full peer review and any attached files.

Reviewer #1: No

Reviewer #2: No

---

## [Author Response · Author response to Decision Letter 0]

23 Aug 2019

The detailed respond is specified in the response to reviewers letter.

---

## [Editor Report · Decision Letter 1]

10 Sep 2019

Time-related immunomodulation by stressors and corticosterone transdermal application in toads

PONE-D-19-16526R1

Dear Dr. Monteiro Titon,

We are pleased to inform you that your manuscript has been judged scientifically suitable for publication and will be formally accepted for publication once it complies with all outstanding technical requirements.

With kind regards,

Marie-Claude Audet

Academic Editor

PLOS ONE
---

## [Editor Report · Acceptance letter]

13 Sep 2019

PONE-D-19-16526R1 

Time-related immunomodulation by stressors and corticosterone transdermal application in toads 

Dear Dr. Titon:

I am pleased to inform you that your manuscript has been deemed suitable for publication in PLOS ONE. Congratulations! Your manuscript is now with our production department. 

With kind regards,

on behalf of

Dr. Marie-Claude Audet 

Academic Editor

PLOS ONE